🔓 | **Open Peer Review** | Bacteriology | Research Article

# Hemolytic activity and antibiotic resistance profiles of *Staphylococcus aureus* isolates from clinical patients

Yen-Hsi Lai,[1] Chih-Chen Kao,[1,2] Min Yi Wong,[1] Tsung-Yu Huang,[2,3] Yu-Hui Lin,[1] Chien-Wei Chen,[2,4] Yao-Kuang Huang[1,2,5,6]

**ABSTRACT** *Staphylococcus aureus* is a common human pathogen that can cause vascular and skin infections, and patients undergoing hemodialysis are particularly susceptible to vascular access infections caused by *S. aureus*. Hemolysins are important virulence factors, and antibiotic resistance poses challenges for treatment. In this study, *S. aureus* isolates were collected from hemodialysis patients with vascular access, such as arteriovenous grafts, tunneled-cuffed catheters, and arteriovenous fistulas, as well as from non-vascular access infection (VAI) patients. The hemolytic phenotype and eight antibiotic susceptibility of these isolates were tested, and PCR was used to detect hemolysin (*hla*, *hlb*, *hld*, and *hlgC/B*) and antibiotic resistance genes (*accA-aphD*, *tetM*, and *tetK*). The results showed that methicillin-resistant *S. aureus* (MRSA) and methicillin-susceptible *S. aureus* (MSSA) isolates exhibited only β- and γ-hemolytic phenotypes. The *hla* and *hld* genes were the most frequently detected hemolysin genes, whereas *hlb* was the least common. Over 80% of both MRSA and MSSA isolates exhibited resistance to ampicillin, with multidrug resistance observed more frequently in MRSA. Distinct antibiotic resistance gene patterns were observed in MRSA and MSSA isolates. Despite these differences in gene patterns, no obvious differences were found between VAI and non-VAI patients, or between MRSA and MSSA isolates. These findings provide a better understanding of the hemolytic characteristics and antibiotic susceptibility of *S. aureus* isolates collected from hemodialysis and non-hemodialysis patients, contributing to more targeted and effective treatment strategies.

**IMPORTANCE** *Staphylococcus aureus* is a common human pathogen, and dialysis patients are at higher risk of infection compared to the general population because this bacterium can colonize medical devices and vascular access catheters. Among its various virulence factors, hemolysins play a crucial role by damaging host cells and helping the bacteria evade immune defenses. The widespread use of antibiotics has led to the emergence of antibiotic-resistant *S. aureus*, especially methicillin-resistant *S. aureus*, further complicating treatment. This study aims to investigate the types of hemolysins, the distribution of hemolysin and antibiotic resistance genes, and antibiotic resistance patterns in *S. aureus* isolates from patients with vascular access-related and non-vascular access infections, providing a reference for infection control and treatment strategies.

**KEYWORDS** *Staphylococcus aureus*, vascular access infections, hemolysis, antibiotic resistance

S taphylococcus aureus is a common pathogenic bacterium in humans. It can cause blood vessels, skin infections, ulcers, and other diseases or deep and systemic infections such as osteomyelitis, endocarditis, pneumonia, and bacteremia. In addition, it can colonize the host without causing any disease (1). About 30% of the population carries this bacterium. Studies have found that dialysis patients are more likely to be infected with *S. aureus* than ordinary patients. The main reason is that *S. aureus* can

**Peer Reviewer** Patrick M. Schlievert, The University of Iowa, Iowa City, Iowa, USA

Address correspondence to Yao-Kuang Huang, yaokuang@gmail.com.

Yen-Hsi Lai and Chih-Chen Kao contributed equally to this article. The author order was determined by drawing straws.

The authors declare no conflict of interest.

See the funding table on p. 10.

*[This article was published on 23 October 2025 with errors in Acknowledgments and Ethics Approval. The errors were corrected in the current version, posted on 14 November 2025.]*

adhere to host cells or matrix surfaces, such as medical devices, leading to cell aggregation and the production of toxins (2, 3). Dialysis equipment thus serves as a good site for bacterial attachment, and skin punctures at vascular access sites and the use of central venous hemodialysis and peritoneal dialysis polymer catheters are also factors that increase the risk of infection (4).

Nearly all *S. aureus* strains secrete diverse enzymes and cytotoxins such as hemolysins, nucleases, proteases, lipases, hyaluronidases, and collagenases and can also produce superantigens (SAgs), a family of potent immunostimulatory exotoxins that suppress host defenses against pathogenic strains. The primary function of these proteins may be to convert local tissues into nutrients needed for bacterial growth (5). Hemolysin plays an important role in the pathogenesis of all diseases caused by *S. aureus* by helping to lyse host cell membranes, disrupt or evade the immune system, and release nutrients for pathogen survival and disease development (6). Hemolysins can be divided into four types including α-, β-, δ-, and γ-hemolysins. α-Hemolysin, also known as alpha-toxin, is a cytotoxic, hemolytic protein that causes extensive damage to host cells including epithelial cells, endothelial cells, erythrocytes, monocytes, and keratinocytes, destroys cell membranes, and induces apoptosis. Encoded by the *hla* gene on the chromosome, it plays a role in many bacterial infections (5, 7). β-Hemolysin significantly affects the function of human immune cells. It is a sphingomyelinase encoded by *hlb*. It breaks down various types of cells according to the content of sphingomyelin, such as lymphocytes, keratinocytes, and neutrophils (6, 8). δ-Hemolysin is a low molecular weight exotoxin, a 26-amino acid peptide toxin produced by certain strains of *S. aureus*. It is encoded by the *hld* gene, which is part of RNAIII, an accessory gene regulator related to the phenolic soluble regulatory proteins (PSM), a family of peptide toxins. This toxin has multiple functions, including lysis of red blood cells, post-phagocytic neutrophils, and other mammalian cells, as well as toxic effects on other bacterial cells (5, 9). Moreover, β-toxin and δ-toxin exhibit strong hemolysis through synergistic effect (10). γ-Hemolysin is a pore-forming toxin (PFT) produced by *S. aureus* that lyses red blood cells in a variety of mammalian species (11). In addition, γ-toxin exerts toxic effects on phagocytes at the site of infection to evade the immune system. Furthermore, this hemolysin is encoded by the *hlg* gene (5, 7).

The invention of antibiotics has led to the emergence of drug-resistant strains, with *S. aureus* being particularly notorious for its resistance to penicillin. Epidemic clones often originate from one or a few successful lineages and have spread globally, and methicillin-resistant *S. aureus* (MRSA) is the most prominent example of these outbreaks (12). Its methicillin resistance is conferred by the *mecA* gene (13). In addition, *S. aureus* also possesses other antibiotic resistance genes, such as aminoglycoside (such as gentamicin) resistance, which is coded by *aacA-aphD* gene, and tetracycline (such as doxycycline and tetracycline) resistance, which is coded by *tetK* and *tetM* genes (14, 15). Drug-resistant strains have emerged, including multidrug-resistant strains that are resistant to three or more classes of antibacterial agents. These strains pose significant challenges for clinical treatment (16, 17).

This study mainly investigates the hemolytic types and hemolytic gene combination distribution of MRSA and methicillin-susceptible *S. aureus* (MSSA) in the vascular access infection (VAI) and non-VAI, as well as the status of antibiotic minimum inhibitory concentration (MIC) and antibiotic resistance genes, to provide reference for future treatment of patients in the renal dialysis pathway.

## RESULTS

### Detection of hemolytic phenotype among the collected *S. aureus* isolates on the sheep blood agar plate (BAP)

Hemolysis on blood agar is classified as β (clear zone, complete lysis), α (greenish or brownish zone, partial lysis), or γ (no lysis). Among the collected *S. aureus* isolates, only β- and γ-hemolytic phenotypes were observed. Even isolates showing only faint hemolysis were classified as β-hemolytic (Fig. 1a), while strongly β-hemolytic isolates

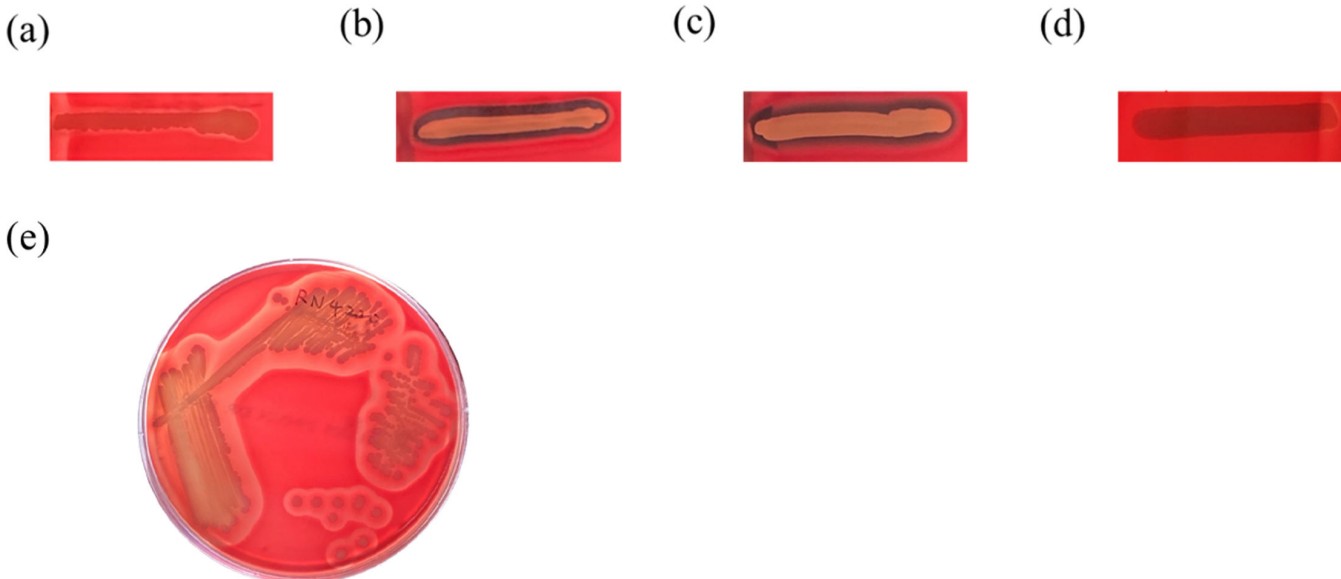

**FIG 1** Hemolysis phenotypes of *S. aureus* isolates from vascular access and non-VAI infections. (a–c) Exhibits the β-hemolysis, (d) displays isolates with γ-hemolysis, and (e) represents the *S. aureus* control strain.

displayed a distinct, transparent zone (Fig. 1b). The control strain produced β-hemolysin, which synergized with δ-hemolysin to enhance hemolysis at the intersection zone, also classified as β-hemolysis (Fig. 1c). γ-Hemolytic isolates showed no visible changes on the BAPs (Fig. 1d). The RN4220 strain, used as a quality control, exhibited only β-hemolytic activity (Fig. 1e).

## Distribution of hemolysis types among MRSA and MSSA isolates

Among the 179 *S. aureus* isolates, 102 were identified as MRSA and 77 as MSSA. The majority of isolates exhibited hemolytic activity on blood agar, with β-hemolysis being more prevalent than γ-hemolysis and accounting for approximately or over 70% in most groups. In contrast to isolates from other infection sources, γ-hemolysis was observed more frequently than β-hemolysis among non-VAI MRSA isolates (Fig. 2).

## Detection of hemolysis activity and hemolysin genes in *S. aureus* isolates

Four hemolysin genes were detected in *S. aureus* isolates, and the prevalence of these hemolysin genes was similar between MRSA and MSSA isolates. The majority of the collected *S. aureus* isolates carried the *hla*, *hld*, and *hlg*C/*hlg*B genes. Notably, the *hld* gene was present in 100% of both MRSA and MSSA isolates, followed by the *hla* gene. Only a small proportion of MRSA isolates and a few TCC-MSSA and MSSA isolates from non-VAI infections harbored the *hlb* gene (Fig. 3).

## Detection of the hemolysin gene patterns in different hemolysis types among *S. aureus* isolates

MRSA and MSSA isolates were further categorized by infection source (VAI and non-VAI), and their hemolytic gene patterns were analyzed according to β-hemolysis and γ-hemolysis phenotypes. The predominant gene pattern was *hla-hld-hlg*C/*hlg*B, which was detected in 50.8% of MRSA isolates with β-hemolysis and 79.1% of MSSA isolates with β-hemolysis. Notably, this gene pattern was present in 100% of both MRSA and MSSA isolates with γ-hemolysis. The second most common gene pattern was *hla-hlb-hld-hlg*C/*hlg*B, accounting for 46% of MRSA isolates with β-hemolysis (Table 1). In all hemolysin gene patterns, the *hld* gene was revealed in the gene heatmap (Fig. 3) and

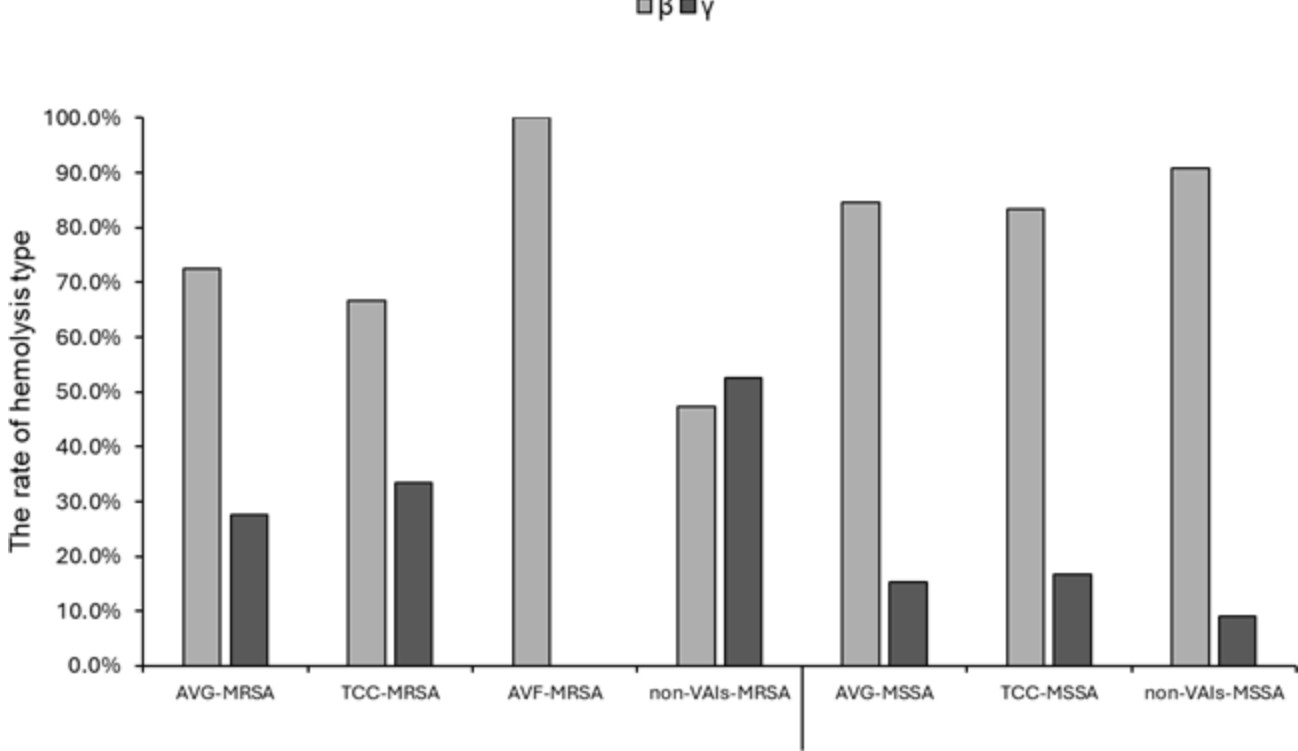

**FIG 2** Hemolysis phenotype profiles of *S. aureus* among collected isolates. The percentage of hemolysis types in MRSA and MSSA isolates.

indicated that this gene was the important factor in hemolysis among the collected *S. aureus* isolates.

## Detection of MICs of antimicrobials and multiple drug resistance among *S. aureus* isolates

Eight antimicrobials were tested, arteriovenous fistula (AVF)-MRSA isolates were only resistant to ampicillin and susceptible to other antibiotics. Ampicillin resistance was observed in 99% (101/102) of MRSA isolates and 87% (67/77) of MSSA isolates. All MRSA and MSSA isolates were susceptible to linezolid, teicoplanin, and vancomycin, except for a few isolates that were not tested. Arteriovenous graft (AVG)-MRSA and tunneled-cuffed catheter (TCC)-MRSA isolates were equally resistant and susceptible to gentamicin. Most MRSA and MSSA isolates were susceptible to doxycycline and rifampicin, though a small number of isolates were resistant to these antimicrobials (Table 2). The proportion of antimicrobial resistance was higher in MRSA isolates than in MSSA isolates; however, for doxycycline and tetracycline resistance, the higher ratio of AVG-MSSA isolates was resistant to these agents compared to AVG-MRSA isolates. This trend was also observed between non-VAI MRSA and MSSA isolates resistant to tetracycline. The prevalence of multidrug resistance (MDR) was higher among MRSA isolates compared to MSSA isolates (Table 3). Furthermore, the MDR ratio of *S. aureus* with AVG infection was less than that of TCC and non-VAI infection. However, statistical analysis revealed no significant difference in MDR rates between MRSA and MSSA isolates (*P* value > 0.05).

## Analysis of the correlation between antimicrobial gene patterns and antimicrobial resistance

There were five antibiotic resistance gene patterns among the collected *S. aureus* isolates. Gene patterns of *accA-aphD* and *accA-aphD/tetM* were prevalent among MRSA isolates (both 85.71%, 18/21), *tetK* (16.46%, 13/79), and *accA-aphD/tetK* (17.72%, 14/79) gene pattern were prevalent in the MSSA isolates. We found that 31.03%

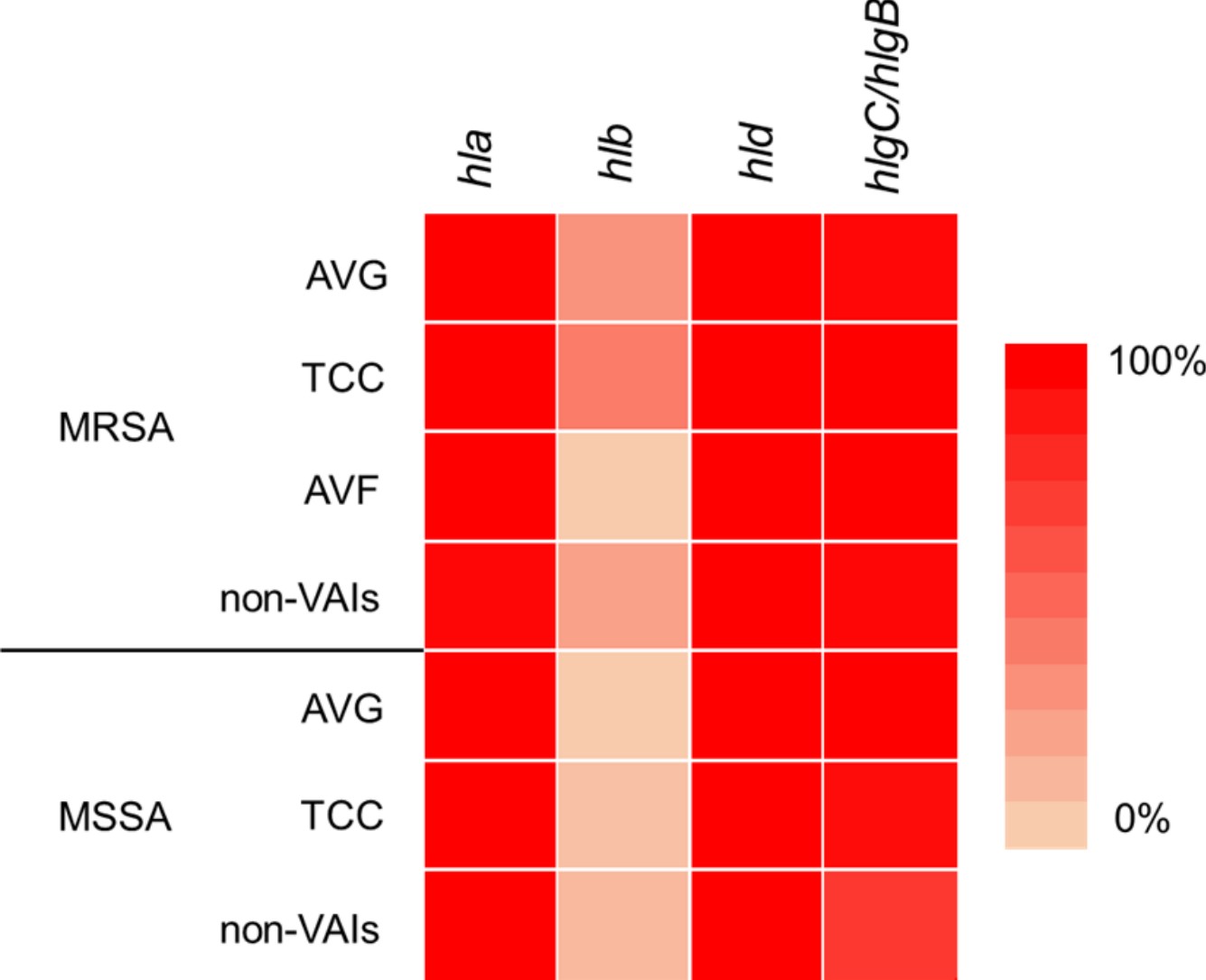

**FIG 3** Heatmap of hemolysis genes in MRSA and MSSA isolates from vascular access infection (VAI) and non-VAI infections. AVG, arteriovenous graft; AVF, arteriovenous fistula; MRSA, methicillin-resistant *Staphylococcus aureus*; MSSA, methicillin-sensitive *Staphylococcus aureus*; TCC, tunneled-cuffed catheter.

(9/29) AVG-MRSA and 18.42% (7/38) non-VAI-MRSA isolates were resistant to gentamicin and frequently carried the *accA-aphD* genes. Additionally, some isolates harbored both *accA-aphD* and *tetM* genes, showing resistance to doxycycline, gentamicin, and tetracycline, particularly among 21.21% (7/33) TCC-MRSA and 21.05% (8/38) non-VAI-MRSA isolates. In contrast, the *tetK* gene was commonly found in MSSA isolates resistant to tetracycline, with the *accA-aphD/tetK* gene pattern conferring resistance to gentamicin and tetracycline (Table 4).

## DISCUSSION

*S. aureus* is a bacterium that possesses a variety of virulence factors that, individually or in combination, enable it to cause a wide range of diseases, including sepsis, infective endocarditis, pneumonia, eye infections, central nervous system infections, skin and soft tissue infections, and bone and joint infections (18). Hemolysin, leukotoxin (Panton-Valentine leukotoxin), and toxic shock syndrome toxin-1 contribute to the destruction of red blood cell membranes, impairment of phagocytic function of leukocytes, and induction of toxic shock syndrome, respectively (19). Among these toxins, hemolysins were classified as α-, β-, γ-, and δ-hemolysins, which are involved in the development of

**TABLE 1** Hemolysin gene patterns in *S. aureus* isolates by hemolysis types

| Type | Category | Hemolysis | Hemolysin gene(s) | No. |
|---|---|---|---|---|
| MRSA (102) | AVG (29) | β (21) | *hla*, *hlb*, *hld*, *hlg*C, *hlg*B | 8 |
| | | | *hla*, *hld*, *hlg*C, *hlg*B | 12 |
| | | | *hla*, *hld* | 1 |
| | | γ (8) | *hla*, *hld*, *hlg*C, *hlg*B | 8 |
| | TCC (33) | β (22) | *hla*, *hlb*, *hld*, *hlg*C, *hlg*B | 13 |
| | | | *hla*, *hld*, *hlg*C, *hlg*B | 9 |
| | | γ (11) | *hla*, *hld*, *hlg*C, *hlg*B | 11 |
| | AVF (2) | β (2) | *hla*, *hld*, *hlg*C, *hlg*B | 2 |
| | non-VAIs (38) | β (18) | *hla*, *hlb*, *hld*, *hlg*C, *hlg*B | 8 |
| | | | *hla*, *hld*, *hlg*C, *hlg*B | 9 |
| | | | *hld* | 1 |
| | | γ (20) | *hla*, *hld*, *hlg*C, *hlg*B | 20 |
| MSSA (77) | AVG (26) | β (22) | *hla*, *hld*, *hlg*C, *hlg*B | 22 |
| | | γ (4) | *hla*, *hld*, *hlg*C, *hlg*B | 4 |
| | TCC (18) | β (15) | *hla*, *hlb*, *hld*, *hlg*C, *hlg*B | 1 |
| | | | *hla*, *hld*, *hlg*C, *hlg*B | 13 |
| | | | *hla*, *hld* | 1 |
| | | γ (3) | *hla*, *hld*, *hlg*C, *hlg*B | 3 |
| | Others (33) | β (30) | *hla*, *hlb*, *hld*, *hlg*C, *hlg*B | 3 |
| | | | *hla*, *hld*, *hlg*C, *hlg*B | 18 |
| | | | *hla*, *hld* | 9 |
| | | γ (3) | *hla*, *hld*, *hlg*C, *hlg*B | 3 |

infectious diseases. α-, γ-, and δ-hemolysins are PFTs that induce lysis of target cells by forming pores in the cell membrane, whereas β-hemolysin (Hlb) is a non-pore-forming toxin (20). α-, β-, δ-, and γ-hemolysins were encoded by *hla*, *hlb*, *hld,* and *hlg* genes, respectively. Furthermore, *S. aureus* is known to exhibit a range of hemolytic phenotypes on blood agar, including α (partial), β (complete), and γ (none) hemolysis, consistent with previous studies (5, 7, 21). In our study, most of the collected *S. aureus* isolates exhibited hemolytic activity, with β-hemolytic isolates accounting for 61.8% of MRSA and 87% of MSSA. In many isolates, an expansion of the hemolytic zone was observed due to the interaction between β-hemolysin and δ-hemolysin, a phenomenon also reported in previous studies (22, 23). Regardless of variations in the presence of other hemolysin genes, *hla* and *hld* were the most frequently carried among *S. aureus* isolates (24–26). In our study, *hla* and *hld* were frequently detected in both MRSA and MSSA isolates, with *hld* being particularly prevalent. Similar findings regarding the widespread distribution of *hld* among staphylococcal strains have been observed in previous studies (27).

The results demonstrated that MRSA exhibited a higher prevalence of MDR compared to MSSA. Both MRSA and MSSA isolates were resistant to ampicillin, doxycycline, gentamicin, and tetracycline, a finding consistent with previous studies (28, 29). A correlation between antibiotic agents and resistance-associated genes was observed. In MRSA isolates, resistance was most frequently observed for gentamicin and doxycycline/gentamicin/tetracycline (both 17.64%, 18/102), corresponding predominantly to the *accA-aphD* and *aacA-aphD/tetM* gene patterns, respectively. In MSSA isolates, resistance to gentamicin/tetracycline (15.19%, 12/79) and tetracycline alone (12.66%, 10/79) was mainly associated with the *aacA-aphD/tetK* and *tetK* gene patterns, respectively. These findings are consistent with previous studies reporting that *accA-aphD* confers gentamicin resistance, while *tetK* and *tetM* are associated with tetracycline and doxycycline resistance (30, 31). Due to the impaired ability of the *tetK* protein to expel certain tetracycline derivatives, the *tetK* gene confers lower resistance to doxycycline (32), which is consistent with previous studies showing increased doxycycline susceptibility in *S. aureus* isolates harboring *tetK* or *tetM* (30, 31, 33). In our study, isolates

**TABLE 2** Minimum inhibitory concentrations of *S. aureus* isolates from various infections[a]

| Drug | | MRSA (102) | | | | MSSA (77) | | |
|------|---|------------|---|---|---|-----------|---|---|
| | | AVF (*n* = 2) (%) | AVG (*n* = 29) (%) | TCC (*n* = 33) (%) | Non-VAI (*n* = 38) (%) | AVG (*n* = 26) (%) | TCC (*n* = 18) (%) | Non-VAI (*n* = 33) (%) |
| AMP | S | 0 (0) | 0 (0) | 0 (0) | 1 (2.63) | 3 (11.54) | 0 (0) | 7 (21.21) |
| | I | 0 (0) | 0 (0) | 0 (0) | 0 (0) | 0 (0) | 0 (0) | 0 (0) |
| | R | 2 (100) | 29 (100) | 33 (100) | 37 (97.37) | 23 (88.46) | 18 (100) | 26 (78.79) |
| DOX | S | 2 (100) | 22 (75.86) | 14 (42.42) | 21 (55.26) | 14 (53.85) | 14 (77.78) | 20 (60.61) |
| | I | 0 (0) | 1 (3.45) | 5 (15.15) | 1 (2.63) | 4 (15.38) | 3 (16.67) | 3 (9.09) |
| | R | 0 (0) | 6 (20.69) | 14 (42.42) | 16 (42.11) | 8 (30.77) | 1 (5.56) | 10 (30.30) |
| GEN | S | 2 (100) | 15 (51.72) | 15 (45.45) | 14 (36.84) | 20 (76.92) | 12 (66.67) | 23 (69.70) |
| | I | 0 (0) | 0 (0) | 0 (0) | 0 (0) | 1 (3.85) | 0 (0) | 1 (3.03) |
| | R | 0 (0) | 14 (48.28) | 18 (54.55) | 24 (63.16) | 5 (19.23) | 6 (33.33) | 9 (27.27) |
| RIF | S | 2 (100) | 26 (89.66) | 32 (96.97) | 30 (78.95) | 24 (92.31) | 16 (88.89) | 32 (96.97) |
| | I | 0 (0) | 2 (6.90) | 0 (0) | 4 (10.53) | 2 (7.69) | 0 (0) | 1 (3.03) |
| | R | 0 (0) | 1 (3.45) | 1 (3.03) | 4 (10.53) | 0 (0) | 2 (11.11) | 0 (0) |
| TET | S | 2 (100) | 22 (75.86) | 10 (30.30) | 18 (47.37) | 15 (57.69) | 11 (61.11) | 12 (36.36) |
| | I | 0 (0) | 0 (0) | 0 (0) | 0 (0) | 0 (0) | 0 (0) | 0 (0) |
| | R | 0 (0) | 7 (24.14) | 23 (69.70) | 20 (52.63) | 11 (42.31) | 7 (38.89) | 21 (63.64) |
| LZD | S | 2 (100) | 29 (100) | 32 (96.97) | 38 (100) | 26 (100) | 18 (100) | 33 (100) |
| | I | 0 (0) | 0 (0) | 0 (0) | 0 (0) | 0 (0) | 0 (0) | 0 (0) |
| | R | 0 (0) | 0 (0) | 0 (0) | 0 (0) | 0 (0) | 0 (0) | 0 (0) |
| TEC | S | 2 (100) | 29 (100) | 33 (100) | 38 (100) | 26 (100) | 18 (100) | 32 (96.97) |
| | I | 0 (0) | 0 (0) | 0 (0) | 0 (0) | 0 (0) | 0 (0) | 0 (0) |
| | R | 0 (0) | 0 (0) | 0 (0) | 0 (0) | 0 (0) | 0 (0) | 0 (0) |
| VAN | S | 2 (100) | 29 (100) | 33 (100) | 38 (100) | 26 (100) | 18 (100) | 33 (100) |
| | I | 0 (0) | 0 (0) | 0 (0) | 0 (0) | 0 (0) | 0 (0) | 0 (0) |
| | R | 0 (0) | 0 (0) | 0 (0) | 0 (0) | 0 (0) | 0 (0) | 0 (0) |

[a]AMP, ampicillin; DOX, doxycycline; GEN, gentamicin; I, intermediate; LZD, linezolid; R, resistant; RIF, rifampicin; S, susceptible; TEC, teicoplanin; TET, tetracycline; VAN, vancomycin.

harboring the *tetK* or *tetM* gene showed resistance to the doxycycline/tetracycline combination, but not to doxycycline alone.

Importantly, isolates from patients with VAI and those without showed largely similar experimental observations, with no notable differences in either hemolytic phenotypes or overall resistance across the eight tested antibiotics. Differences between MRSA and MSSA were mostly reflected in resistance gene patterns rather than in overall resistance.

Limitations of this study include its single-center design and relatively small sample size, which may affect generalizability. Nevertheless, these results provide insights into the hemolytic types and antibiotic resistance profiles of MRSA and MSSA isolates obtained from VAI and non-VAI patient specimens, supporting more targeted clinical management strategies.

## MATERIALS AND METHODS

### Bacterial isolates collection and identification

This research was conducted at Chiayi Chang Gung Memorial Hospital in Chiayi, Taiwan. A total of 179 *S. aureus* isolates were collected from patients with vascular access infections, including AVFs, prosthetic AVGs, TCCs, and other infections such as arthritis and pneumonia, between November 2013 and November 2023. The bacterial isolates were obtained from various sources, including blood, Hickman catheter tips, pus, tissue, wounds, and pleural effusion, and were cultured on BAPs. The isolates were identified using standard biochemical tests, including catalase and coagulase tests before 2019. Matrix-assisted laser desorption/ionization time-of-flight mass spectrometry (MALDI-TOF) was used for identification after 2019. For routine culturing, the isolates

**TABLE 3** The multiple drug resistance (MDR) of *S. aureus* isolates from different infections

| MRSA (102) | No. (%) | MSSA (77) | No. (%) | *P* value |
|---|---|---|---|---|
| AVF (2) | 0 (0) | AVF (0) | 0 (0) | N/A[a] |
| AVG (29) | 6 (20.7) | AVG (26) | 4 (15.4) | 0.612 |
| TCC (33) | 16 (48.5) | TCC (18) | 5 (27.8) | 0.151 |
| Non-VAI (38) | 16 (42.1) | Non-VAI (33) | 10 (30.3) | 0.303 |

[a]"NA" indicates not available.

were grown on tryptic soy agar and in tryptic soy broth under laboratory conditions. All isolates were stored in 15% glycerol stocks at −80°C.

## Hemolysis phenotype analysis on the sheep BAP

Streak the center of the sheep BAP (Becton Dickinson, USA, trypticase soy agar with 5% sheep blood) with the RN4220 *S. aureus* strain that produces only beta-hemolysis. Test strains were streaked perpendicular to, but not touching, the central stripe, and the plates were incubated at 37°C for 24 h. After incubation, the hemolysis of the *S. aureus* isolates was observed, and the phenotypes on the SBA plates were recorded (7, 23, 34). α-Hemolysis: a greenish halo forms around the colonies, reflecting incomplete hemolysis. β-Hemolysis: cultures exhibit a translucent halo around and beneath the colonies, representing complete hemolysis. γ-Hemolysis: no noticeable zones are present around the colonies. Furthermore, the β-hemolysin produced by the control strain synergized with δ-hemolysin, resulting in enhanced hemolysis at the vertical intersection zone. This enhanced hemolysis was also classified as β-hemolysis (23, 35).

## Hemolysin genes detection

In the hemolysis detection of bacterial isolates, α-, β-, δ-, and γ-hemolysins are encoded by *hla*, *hlb*, *hld*, *and hlg* genes, respectively. These four hemolysin genes, including *hla*, *hlb*, *hld* (36), and *hlgC/hlgB* (37) genes, were amplified utilizing the polymerase chain reaction (PCR) in previous studies.

## MIC testing

This detection was conducted in accordance with a previous study (38). Bacterial isolates to be tested were streaked onto agar plates without inhibitors to obtain single colonies. The plates were incubated at 37°C for 18–24 h. For each isolate, three to five morphologically similar colonies were selected and transferred into a tube containing Mueller–Hinton broth for agitation (shocking) for 1–2 h. A 96-well sterile microtiter plate was labeled with the respective antibiotic concentrations, designating one row for each test, accommodating up to 10 different dilutions. Into columns 2–10, 50 µL of broth was added, while 100 µL of broth was added to the sterility control well (column 12) and 50 µL to the growth control well (column 11). Subsequently, add 100 µL of the antibiotic solution to column 1, then perform a twofold serial dilution of each antibiotic across the respective wells. For each bacterial isolate, the bacterial suspension was adjusted to a concentration of $1 \times 10^8$ CFU/mL by vortexing (based on McFarland standard 0.5 or a optical density [OD] at 625 nm of 0.08–0.13) and diluted 1:100. Each well containing the antibiotic solution, along with the growth control well, was inoculated with 50 µL of the prepared bacterial suspension, achieving a final inoculum of $5 \times 10^5$ CFU/mL. The microtiter plate was incubated at 37°C for 16–20 h. Overnight monitoring of the OD at 600 nm was performed using a 96-well plate reader. The MIC was defined as the lowest antibiotic concentration at which there was no visible bacterial growth (no turbidity observed with the naked eye) after 16–20 h of incubation. MIC standards and antimicrobial quality controls adhered to the guidelines of the Clinical and Laboratory Standards Institute (39).

**TABLE 4** Relationship between antibiotic gene patterns and antibiotic resistance[a]

| Type | Category | Genes | DOX | GEN | TET | DOX/TET | GEN/TET | DOX/GEN/TET |
|---|---|---|---|---|---|---|---|---|
| MRSA (102) | AVG (29) | aacA-aphD (10) | | 9 | | | | |
| | | aacA-aphD/tetK (2) | | | | | 1 | 1 |
| | | aacA-aphD/tetM (3) | | | | | | 3 |
| | TCC (33) | aacA-aphD (5) | | 2 | | | | 3 |
| | | aacA-aphD/tetK (6) | | | 2 | | 2 | 1 |
| | | aacA-aphD/tetM (8) | | | | | 1 | 7 |
| | | aacA-aphD/tetK/tetM (1) | | | | | | 1 |
| | | tetK (5) | | | 2 | 2 | 1 | |
| | AVF (2) | –[b] | – | – | – | – | – | – |
| | Non-VAI (38) | aacA-aphD (8) | | 7 | | | 1 | |
| | | aacA-aphD/tetK (2) | | | | | 2 | |
| | | aacA-aphD/tetM (9) | | | | 1 | | 8 |
| | | aacA-aphD/tetK/ tetM (3) | | | | | | 3 |
| | | tetK (3) | | | 2 | | | 1 |
| MSSA (77) | AVG (26) | aacA-aphD/tetK (2) | | | | | 2 | |
| | | aacA-aphD/tetM (2) | | | | | | 2 |
| | | tetK (6) | | | 3 | 3 | | |
| | TCC (18) | aacA-aphD (1) | | 1 | | | | |
| | | aacA-aphD/tetK (4) | | | | | 3 | 1 |
| | | tetK (2) | | | 2 | | | |
| | Non-VAI (33) | aacA-aphD (2) | | | | 1 | | |
| | | aacA-aphD/tetK (8) | | | | 1 | 7 | |
| | | tetK (5) | | | 5 | | | |

[a]DOX, doxycycline; GEN, gentamicin; TET, tetracycline.
[b]"–" indicates that no detection was observed in the AVF samples.

## Antimicrobial resistance gene detection

PCR was utilized to identify the isolates as MRSA, which exhibited oxacillin resistance and tested positive for the *mecA* gene using primers described previously (40). Additionally, the *accA-aphD*, *tetK*, and *tetM* genes, encoding aminoglycoside and tetracycline resistance, respectively, were amplified as described by a previous study (14).

## Statistical analysis

The difference in the rate of MDR between MRSA and MSSA isolates was analyzed using the chi-square test in IBM SPSS Statistics for Windows, Version 22.0 (Armonk, NY: IBM Corp.). A *P* value of less than 0.05 was considered statistically significant.

## Conclusion

This study investigated *S. aureus* isolates from both dialysis and non-dialysis patients. The isolates were classified into MRSA and MSSA, and their characteristics were analyzed according to VAI and non-VAI status. Isolates from VAI and non-VAI patients showed generally similar hemolysis patterns and antibiotic resistance profiles, suggesting comparable phenotypic traits regardless of dialysis status. MRSA isolates exhibited a higher proportion of MDR and their resistance gene profiles differed slightly in composition from those of MSSA. These findings highlight that while dialysis status may not strongly influence *S. aureus* phenotypes, the distinction between MRSA and MSSA remains critical. The phenotypic and genetic investigation of *S. aureus* isolates is crucial for infection management and the development of targeted treatment strategies.

## ACKNOWLEDGMENTS

We thank the Department of Laboratory Medicine at Chang Gung Memorial Hospital, Chiayi, Taiwan, for their assistance in collecting the bacterial isolates.

This study was funded by the Chang Gung Memorial Hospital, Chiayi, Taiwan (grant number: CMRPG6M0101-3, CMRPG6N0241-2, CMRPG6P0251-2, NMRPG6P0071, NMRPG6Q0091).

Y.-H. Lai: data curation, formal analysis, investigation, writing—original draft, writing—review and editing. C.-C.K.: conceptualization, funding acquisition, project administration, writing—original draft, writing—review and editing. M.Y.W.: data curation, formal analysis, validation, writing—review and editing. T.-Y.H.: funding acquisition, methodology, resources, writing—review and editing. Y.-H. Lin: investigation, validation, writing—review and editing. C.-W.C.: investigation, validation, writing—review and editing. Y.-K.H.: conceptualization, funding acquisition, methodology, project administration, resources, supervision, writing—original draft, writing—review and editing. All the authors have read and agreed to the published version of the manuscript.

## AUTHOR AFFILIATIONS

[1]Division of Thoracic and Cardiovascular Surgery, Chiayi Chang Gung Memorial Hospital, Puzi, Taiwan
[2]College of Medicine, Chang Gung University, Taoyuan, Taiwan
[3]Division of Infectious Diseases, Department of Internal Medicine, Chiayi Chang Gung Memorial Hospital, Puzi, Taiwan
[4]Department of Diagnostic Radiology, Chiayi Chang Gung Memorial Hospital, Puzi, Taiwan
[5]Division of Cardiovascular Surgery, New Taipei Municipal TuCheng Hospital, New Taipei, Taiwan
[6]Division of Thoracic and Cardiovascular Surgery, Chiayi Hospital, Ministry of Health and Welfare, Taipei City, Taiwan

## AUTHOR ORCIDs

Yao-Kuang Huang  http://orcid.org/0000-0003-2699-2207

## FUNDING

| Funder | Grant(s) | Author(s) |
| --- | --- | --- |
| Chiayi Chang Gung Memorial Hospital | CMRPG6M0101, CMRPG6M0102, CMRPG6M0103, CMRPG6N0241-2, CMRPG6P0251-2, NMRPG6P0071, NMRPG6Q0091 | Yen-Hsi Lai |
| | | Chih-Chen Kao |
| | | Yao-Kuang Huang |

## AUTHOR CONTRIBUTIONS

Yen-Hsi Lai, Data curation, Formal analysis, Investigation, Writing – original draft, Writing – review and editing | Chih-Chen Kao, Conceptualization, Funding acquisition, Project administration, Writing – review and editing | Min Yi Wong, Data curation, Formal analysis, Investigation, Validation, Writing – review and editing | Tsung-Yu Huang, Funding acquisition, Methodology, Resources, Writing – review and editing | Yu-Hui Lin, Investigation, Validation, Writing – review and editing | Chien-Wei Chen, Investigation, Validation, Writing – review and editing | Yao-Kuang Huang, Conceptualization, Funding acquisition, Methodology, Project administration, Resources, Supervision, Writing – original draft, Writing – review and editing

## DATA AVAILABILITY

The authors declare that the experimental data presented in this article are available upon request.

## ETHICS APPROVAL

This study was approved by the Institutional Review Board (IRB) of Chang Gung Memorial Hospital (IRB101-4188B, IRB104-8482B, IRB201801166B0, IRB201901354B0, and IRB 202200511B0) and was conducted in accordance with the approved guidelines.

## ADDITIONAL FILES

The following material is available online.

### Open Peer Review

**PEER REVIEW HISTORY (review-history.pdf).** An accounting of the reviewer comments and feedback.

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
