## [Reviewer comments · Microbiology Spectrum]

Microbiology Spectrum

Hemolytic activity and antibiotic resistance profiles of *Staphylococcus aureus* isolates from clinical patients

Yen-Hsi Lai, Chih-Chen Kao, Min Yi Wong, Tsung-Yu Huang, Yu-Hui Lin, Chien-Wei Chen, and Yao-Kuang Huang

Corresponding Author(s): Yao-Kuang Huang, Chiayi Chang Gung Memorial Hospital

Review Timeline:

Submission Date:	July 10, 2025
Editorial Decision:	August 13, 2025
Revision Received:	August 25, 2025
Editorial Decision:	September 15, 2025
Revision Received:	September 17, 2025
Accepted:	September 26, 2025

Editor: Kunyan Zhang

Reviewer(s): Disclosure of reviewer identity is with reference to reviewer comments included in decision letter(s). The following individuals involved in review of your submission have agreed to reveal their identity: Patrick M Schlievert (Reviewer #1)

Transaction Report:

DOI: <https://doi.org/10.1128/spectrum.02074-25>

Re: Spectrum02074-25 (**Hemolytic activity and antibiotic resistance profiles of *Staphylococcus aureus* isolates from clinical patients**)

Dear Dr. Yao-Kuang Huang:

Thank you for the privilege of reviewing your work. Below you will find my comments, instructions from the Spectrum editorial office, and the reviewer comments.

Revision Guidelines

Sincerely,
Kunyan Zhang
Editor
Microbiology Spectrum

Reviewer #1 (Comments for the Author):

The is a potentially interesting manuscript that studies the hemolysin production and antibiotic resistance patterns of vascular access infections versus non-vascular access infections. The authors' use of English needs a lot of work to allow the paper to be reviewed carefully. The section on antibiotic susceptibility is OK, but the hemolysis and hemolysin parts are confusing. Specifically:

1. There is a standard definition of hemolysis, as alpha (greening of blood agar, beta (clearing), and gamma (no hemolysis). I am not sure in the figures 2 and 3 what exactly are the authors saying. They should use the standard definition of hemolysis, and then as they have done used the standard definitions of the specific hemolysins.

There are many language problems: Line 77- what does toxin-causing toxins mean? Line 82- should mention superantigens as part of the toxin profile. line 89- lethal protein only refers to lethality to various cell types, not humans. None of the cytotoxins are particularly lethal to humans. Line 109_ I don't have any idea what is being said. Table 2.1 and 2.2 appear to be distinct Tables and should be re-numbered as such. Resistance to and resistant to are different. The authors should say the organisms are resistant to some antibiotic, not the organisms are resistance to some antibiotic. line 236 non-purulent(toxin-mediated) is incorrect. All *S. aureus* strains make at least one hemolysin and at least one superantigen. This has been known for many years. Schlievert and Leung showed this. *S. aureus* should be italicized.

Reviewer #2 (Comments for the Author):

Lai et al. determine hemolysis caused by *Staphylococcus aureus* isolated from hemodialysis patients and correlate this hemolysis with hemolysins and antibiotic resistance. The study has potential to provide useful data. However, the term hemolysis is used incorrectly, and the manuscript text needs considerable editing for English grammar and syntax. Notably, some key words in selected sentences seem inadvertently placed. Specific comments are provided below.

1. The main conclusions (no significant difference between dialysis and non-dialysis patients, and MRSA versus MSSA) should be presented in the Abstract and discussed in more detail in the Discussion section. The Conclusion section reveals the findings with dialysis versus non-dialysis, but this key point of the study needs more discussion.

2. Bacterial hemolysis is complete (beta-hemolysis), partial (alpha-hemolysis), or none (gamma-hemolysis). There is no delta-hemolysis (as termed in this study). There can be complete or partial hemolysis caused by delta-hemolysin, but this is simply hemolysis. Please revise the manuscript and display items to follow standard convention. For *S. aureus*, this methodology and nomenclature was published many decades ago (e.g., see S.D. Elek and E. Levy, *J. Path. Bact.*, 1950, 62: 541-54; G.A. Hebert and G.A. Hancock, *J. Clin. Microbiol.*, 1985, 22: 409-15).

3. Please revise the text throughout, as syntax is incorrect in many areas or there are simply errors. A few examples are provided below.

Lines 76-77: "...staphylococcus aureus will attach to host cells or matrix surfaces such as medical equipment to produce cell aggregation and toxin-causing toxins [2] [3]."

Lines 108-109: "One or a few successfully cloned strains cause many antibiotic-resistant strains or infections."

Lines 115-118: "When antibiotics are used to treat bacterial infections, in addition to producing drug-resistant strains, there are even multi-drug-resistant strains, which are resistant to three or more antibacterial drug classes."

August 25, 2025

Dear Editor and Reviewers

We are submitting our revised manuscript (Manuscript ID: Spectrum02074-25) entitled "Hemolytic activity and antibiotic resistance profiles of *Staphylococcus aureus* isolates from clinical patients" for consideration of "Microbiology Spectrum" after revise. Thank you again for granting us the privilege to revise the paper. We have specifically responded to the reviewers' questions and criticisms point-by-point as follows and added them into this revised version.

Reviewer 1

[Comment 1]

There is a standard definition of hemolysis, as alpha (greening of blood agar, beta (clearing), and gamma (no hemolysis). I am not sure in the figures 2 and 3 what exactly are the authors saying. They should use the standard definition of hemolysis, and then as they have done used the standard defintions of the specific hemolysins.

[Answer 1]

Thank you for your suggestion, we adjust the sentences.

(Line 127-135) [Hemolysis on blood agar is classified as β (clear zone, complete lysis), α (greenish or brownish zone, partial lysis), or γ (no lysis). Among the collected *S. aureus* isolates, only β - and γ -hemolytic phenotypes were observed. Even isolates showing only faint hemolysis were classified as β -hemolytic (Figure 1a), while strongly β -hemolytic isolates displayed a distinct, transparent zone (Figure 1b). The control strain produced β -hemolysin, which synergized with δ -hemolysin to enhance hemolysis at the intersection zone, also classified as β -hemolysis (Figure 1c). γ -hemolytic isolates showed no visible changes on the blood agar plates (Figure 1d). The RN4220 strain, used as a quality control, exhibited only β -hemolytic activity (Figure 1e).]

[Comment 2]

Line 77- what does toxin-causing toxins mean?

[Answer 2]

Thank you for your feedback. We have revised the parts that were difficult to interpret.

(Line 74-76). [The main reason is that *Staphylococcus aureus* can adhere to host cells or matrix surfaces, such as medical devices, leading to cell aggregation and the production of toxins.]

[Comment 3]

Line 82- should mention superantigens as part of the toxin profile.

[Answer 3]

Thank you for your suggestion. We have incorporated the information on superantigens into the manuscript.

(Line 82). [Nearly all *S. aureus* strains secrete diverse enzymes and cytotoxins such as hemolysins, nucleases, proteases, lipases, hyaluronidases, and collagenases, and can also produce superantigens (SAGs), a family of potent immunostimulatory exotoxins that suppress host defenses against pathogenic strains.]

[Comment 4]

line 89- lethal protein only refers to lethality to various cell types, not humans. None of the cytotoxins are particularly lethal to humans.

[Answer 4]

Thank you for your reminder. We have revised this section.

(Line 89). [Hemolysins can be divided into four types including α , β , δ and γ hemolysins. α -hemolysin, also known as alpha-toxin, is a cytotoxic, hemolytic protein that causes extensive damage to host cells including epithelial cells, endothelial cells, erythrocytes, monocytes, and keratinocytes, destroys cell membranes, and induces apoptosis.]

[Comment 5]

Line 109_ I don't have any idea what is being said. Table 2.1 and 2.2 appear to be distinct Tables and should be re-numbered as such. Resistance to and resistant to are different. The authors should say the organisms are resistant to some antibiotic, not the organisms are resistance to some antibiotic.

[Answer 5]

Thank you for your suggestions. We have corrected the unclear phrasing (Line 107-111), renumbered Table 2-1 and 2-2 as Table 2 and Table 3 and changed all instances of “resistance” to “resistant.” (Line 187-202).

[Comment 6]

line 236 non-purulent(toxin-mediated) is incorrect. All *S. aureus* strains make at least one hemolysin and at least one superantigen. This has been known for many years. Schlievert and Leung showed this. *S. aureus* should be italicized.

[Answer 6]

Thank you for your reminder. We have revised this paragraph.

(Line 231-234). [*Staphylococcus aureus* is a bacterium that possesses a variety of virulence factors that, individually or in combination, enable it to cause a wide range of diseases, including sepsis, infective endocarditis, pneumonia, eye infections, central nervous system infections, skin and soft tissue infections, and bone and joint infections.]

Reviewer 2

[Comment 1]

The main conclusions (no significant difference between dialysis and non-dialysis patients, and MRSA versus MSSA) should be presented in the Abstract and discussed in more detail in the Discussion section. The Conclusion section reveals the findings with dialysis versus non-dialysis, but this key point of the study needs more discussion.

[Answer 1]

Thank you for your valuable suggestions. We have revised the abstract, discussion, and conclusion sections

[Comment 2]

Bacterial hemolysis is complete (beta-hemolysis), partial (alpha-hemolysis), or none (gamma-hemolysis). There is no delta-hemolysis (as termed in this study). There can be complete or partial hemolysis caused by delta-hemolysin, but this is simply hemolysis. Please revise the manuscript and display items to follow standard convention. For *S. aureus*, this methodology and nomenclature was published many

decades ago (e.g., see S.D. Elek and E. Levy, J. Path. Bact., 1950, 62: 541-54; G.A. Hebert and G.A. Hancock, J. Clin. Microbiol., 1985, 22: 409-15).

[Answer 2]

Thank you for the reminder. I have revised the description of bacterial hemolysis (line 127-151).

[Comment3]

Please revise the text throughout, as syntax is incorrect in many areas or there are simply errors. A few examples are provided below.

[Answer 3]

Thank you for your valuable comments. We have made revisions to improve the text.

[Comment 4]

Lines 76-77: "...staphylococcus aureus will attach to host cells or matrix surfaces such as medical equipment to produce cell aggregation and toxin-causing toxins [2] [3]."

[Answer 4]

Thank you for your reminder. We have revised this paragraph.

(Line 76-77). [The main reason is that *S. aureus* can adhere to host cells or matrix surfaces, such as medical devices, leading to cell aggregation and the production of toxins.]

[Comment 5]

Lines 108-109: "One or a few successfully cloned strains cause many antibiotic-resistant strains or infections."

[Answer 5]

Thank you for your reminder. We have revised this paragraph.

(Line 107-111). [The invention of antibiotics has led to the emergence of drug-resistant strains, with *S. aureus* being particularly notorious for its resistance to penicillin. Epidemic clones often originate from one or a few successful lineages and have spread globally, and methicillin-resistant *S. aureus* (MRSA) is the most prominent example of these outbreaks.]

[Comment 6]

Lines 115-118: "When antibiotics are used to treat bacterial infections, in addition to producing drug-resistant strains, there are even multi-drug-resistant strains, which are resistant to three or more antibacterial drug classes."

[Answer 6]

Thank you for your reminder. We have revised this paragraph.

(Line 115-117). [Drug-resistant strains have emerged, including multi-drug-resistant strains that are resistant to three or more classes of antibacterial agents. These strains pose significant challenges for clinical treatment.]

Thank you for the informative and careful reviewing our manuscript. We learn a lot during the revision of this article. Please contact me at the following address for additional information.

Sincerely,

Yao-Kuang Huang, MD, PhD

Division of Thoracic and Cardiovascular Surgery

Chia-Yi Chang Gung Memorial Hospital, Putz, Taiwan.

Fax: 886-975368209

E-mail: yaokuang@gmail.com

Re: Spectrum02074-25R1 (**Hemolytic activity and antibiotic resistance profiles of *Staphylococcus aureus* isolates from clinical patients**)

Dear Dr. Yao-Kuang Huang:

Thank you for the privilege of reviewing your work. Below you will find my comments, instructions from the Spectrum editorial office, and the reviewer comments.

Please return the manuscript within 30 days; if you cannot complete the modification within this time period, please contact me. If you do not wish to modify the manuscript and prefer to submit it to another journal, notify me immediately so that the manuscript may be formally withdrawn from consideration by Spectrum.

Revision Guidelines

Sincerely,
Kunyan Zhang
Editor
Microbiology Spectrum

Reviewer #1 (Comments for the Author):

This manuscript addresses hemolytic and antibiotic resistance patterns in *Staphylococcus aureus* isolates. The authors have revised the manuscript, clarifying difficult wording and tables. I have no suggestions.

Reviewer #2 (Comments for the Author):

The article is much improved. Thank you. There are a few minor points that need clarification or editing.

Line 118: replace "understands" with "investigates".

Table 2: please define S, I, and R in the footnote.

Line 219: I assume "accApatternsnd" is a mistake and should be deleted?

Lines 221-222: Consider deleting "In MRSA isolates were revealed that" with "We found that...", because it is clear from the terms AVG-MRSA and non-VAIs-MRSA that these are MRSA isolates.

Line 242. I recommend removing the sentence, "It is essential for the pathogenicity of *S. aureus* [19]." The text is ambiguous (to what does "It" refer?). If it refers to Hla, PVL, and TSST-1 then it is not accurate or cited accurately. Hla is a significant contributor to the full pathogenic potential, but not necessarily essential because CC30/ST36 strains that lacked the ability to produce Hla (b/c of a stop codon) caused numerous healthcare associated infections. Many pathogenic strains lack PVL and/or TSST-1. They are not essential in those lineages.

Line 260: Revise the sentence "We found that MRSA had higher antibiotic resistance than MSSA." This seems self-evident because of beta-lactam resistance. I think the intended meaning is multidrug resistance above and beyond beta-lactam resistance.

Lines 314-317. Please revise for clarity. Delta hemolytic and beta/delta-hemolysis are confusing. This may have been unrevised from the original manuscript?

September 16, 2025

Dear Editor and Reviewers

We are submitting our revised manuscript (Manuscript ID: Spectrum02074-25) entitled "Hemolytic activity and antibiotic resistance profiles of *Staphylococcus aureus* isolates from clinical patients" for consideration of "Microbiology Spectrum" after revise. Thank you again for granting us the privilege to revise the paper. We have specifically responded to the reviewers' questions and criticisms point-by-point as follows and added them into this revised version.

Reviewer 1

[Comment 1]

This manuscript addresses hemolytic and antibiotic resistance patterns in *Staphylococcus aureus* isolates. The authors have revised the manuscript, clarifying difficult wording and tables. I have no suggestions.

[Answer 1]

Thank you for the update. We appreciate the reviewer's positive feedback and are glad to know that no further revisions are needed.

Reviewer 2

[Comment 1]

Line 118: replace "understands" with "investigates".

[Answer 1]

Thank you for your reminder. We have replaced this word.

(Line 118) [This study mainly understands the hemolytic types and hemolytic gene combination distribution of MRSA and MSSA...]

[Comment 2]

Table 2: please define S, I, and R in the footnote.

[Answer 2]

Thank you for your reminder. We have added the annotation. (Line 207)

[Comment 3]

Line 219: I assume "accApatternsnd" is a mistake and should be deleted?

[Answer 3]

Thank you for your reminder. We have made corrections.

(Line 216) [Gene patterns of *accA-aphD* and *accA-aphD/tetM* were prevalent among MRSA isolates...]

[Comment 4]

Lines 221-222: Consider deleting "In MRSA isolates were revealed that" with "We found that...", because it is clear from the terms AVG-MRSA and non-VAIs-MRSA that these are MRSA isolates.

[Answer 4]

Thank you for your suggestion. We have made the modification.

(Line 218) [We found that 31.03% (9/29) AVG-MRSA and 18.42% (7/38) non-VAIs-MRSA isolates were resistant...]

[Comment 5]

Line 242. I recommend removing the sentence, "It is essential for the pathogenicity of *S. aureus* [19]." The text is ambiguous (to what does "It" refer?). If it refers to Hla, PVL, and TSST-1 then it is not accurate or cited accurately. Hla is a significant contributor to the full pathogenic potential, but not necessarily essential because CC30/ST36 strains that lacked the ability to produce Hla (b/c of a stop codon) caused numerous healthcare associated infections. Many pathogenic strains lack PVL and/or TSST-1. They are not essential in those lineages.

[Answer 5]

Thank you for your suggestion. We have revised the sentence.

(Line 239) [Hemolysin, leukotoxin (Panton-Valentine leukotoxin, PVL) and toxic shock syndrome toxin-1 (TSST-1) contribute to the destruction of red blood cell membranes, impairment of phagocytic function of leukocytes and induction of toxic shock syndrome, respectively [19]. Among these toxins, hemolysins were classified as

α -, β -, γ - and δ -hemolysins...]

[Comment 6]

Line 260: Revise the sentence "We found that MRSA had higher antibiotic resistance than MSSA." This seems self-evident because of beta-lactam resistance. I think the intended meaning is multidrug resistance above and beyond beta-lactam resistance.

[Answer 6]

We appreciate your reminder and have revised the sentence.

(Line 256) [The results demonstrated that MRSA exhibited a higher prevalence of multidrug resistance compared to MSSA.]

[Comment 7]

Lines 314-317. Please revise for clarity. Delta hemolytic and beta/delta-hemolysis are confusing. This may have been unrevised from the original manuscript?

[Answer 7]

Thank you for your reminder. We have revised this section.

(Line 305-311) [Alpha-hemolysis: A greenish halo forms around the colonies, reflecting incomplete hemolysis. Beta-hemolysis: Cultures exhibit a translucent halo around and beneath the colonies, representing complete hemolysis...]

Thank you for the informative and careful review of our manuscript. We have gained valuable insights throughout the revision process. Please contact me at the following address if any further information is needed.

Sincerely,

Yao-Kuang Huang, MD, PhD

Division of Thoracic and Cardiovascular Surgery

Chia-Yi Chang Gung Memorial Hospital, Putz, Taiwan.

Fax: 886-975368209

E-mail: yaokuang@gmail.com

Re: Spectrum02074-25R2 (**Hemolytic activity and antibiotic resistance profiles of *Staphylococcus aureus* isolates from clinical patients**)

Dear Dr. Yao-Kuang Huang:

Your manuscript has been accepted, and I am forwarding it to the ASM production staff for publication. Your paper will first be checked to make sure all elements meet the technical requirements. ASM staff will contact you if anything needs to be revised before copyediting and production can begin. Otherwise, you will be notified when your proofs are ready to be viewed.

Sincerely,
Kunyan Zhang
Editor
Microbiology Spectrum